# Chemical Compositions of Essential Oil Extracted from Eight Thyme Species and Potential Biological Functions

**DOI:** 10.3390/plants12244164

**Published:** 2023-12-15

**Authors:** Yanmei Dong, Ziling Wei, Rui Yang, Yanan Zhang, Meiyu Sun, Hongtong Bai, Meiling Mo, Chunlei Yao, Hui Li, Lei Shi

**Affiliations:** 1State Key Laboratory of Plant Diversity and Specialty Crops, Institute of Botany, Chinese Academy of Sciences, Beijing 100093, China; 15501877836@163.com (Y.D.); sylvesterw@163.com (Z.W.); yangrui1993@ibcas.ac.cn (R.Y.); zhangyanan@ibcas.ac.cn (Y.Z.); sunmeiyu@ibcas.ac.cn (M.S.); bai13910086470@163.com (H.B.); 2China National Botanical Garden, Beijing 100093, China; 3University of Chinese Academy of Sciences, Beijing 100049, China; 4Sinno Cosmetics Co., Ltd., Zhongshan 528451, China; emilie@sinnolab.com (M.M.); ray_yao@sinnolab.com (C.Y.)

**Keywords:** thyme, essential oil, chemical type, antioxidant, multivariate statistical analysis

## Abstract

*Thymus* is an herbaceous perennial or subshrub of the Lamiaceae family and is widely distributed worldwide. Essential oils extracted from *thymus* have attracted much attention, owing to their potential biological functions. Here, we evaluated the chemical compositions of eight thyme essential oils (TEOs) using gas chromatography mass spectrometry and assessed their antioxidant activity. The results showed that (1) the main components in eight TEOs were monoterpene hydrocarbons and oxygenated monoterpenes (84.26–92.84%), and the chemical compositions of the TEOs were affected by the specie factor; (2) eight TEOs could be divided into three groups (thymol-, geraniol-, and nerol acetate-types), and thymol was the main type; (3) eight TEOs had some common compounds, such as thymol and p-cymene, which were the main components in seven TEOs; (4) eight TEOs had antioxidant activity, and *Thymus pulegioides*, *Thymus thracicus,* and *Thymus serpyllum* EOs had stronger antioxidant activity than vitamin E (0.07–0.27 fold) at a concentration of 1 mg/mL, while *Thymus quinquecostatus* and *Thymus longicaulis* EOs had relatively weak antioxidant activity. In addition, three chemical type standards were used to evaluate potential roles in antibacterial and tumor therapy. The results showed that thymol had strong antibacterial activity against the growth of *Escherichia coli* and *Staphylococcus aureus*, and antimigratory activity for A549 cell. Overall, our results can provide a theoretical basis for further exploring the function of natural products from thyme essential oils.

## 1. Introduction

*Thymus* is an herbaceous perennial or subshrub of the Lamiaceae family and is widely distributed worldwide. Thyme species are important aromatic and medicinal plants that have been used as a traditional medicine for thousands of years in the Mediterranean basin [1]. Thyme essential oils (TEOs) have strong anti-inflammatory, antioxidant, antimicrobial, and antifungal functions [2,3,4,5]. Cutillas et al. found through the evaluation of the antioxidant capacities of eight thymes that their activity were mainly due to thymol and linalool [6]. TEOs have complex components, and the combination of plant antimicrobials can show synergistic or additive effects [7]. The Chinese native thyme species *Thymus quinquecostatus* is also widely used in folk medicine for the treatment of stroke, cold, dyspepsia, toothache, acute gastroenteritis, hypertension, chronic eczema, and other diseases [1]. In 1986, the World Health Organization identified thyme as an important medicinal plant and included it in the list of medicinal plants. In the *European Pharmacopoeia* (published in 2002), it was stipulated that the sum of thymol and carvacrol contents must be greater than 40% of the total essential oil contents from dried plant materials in order to be used for medicinal purposes [8]. Most essential oils are generally recognized as safe at a certain concentration and are commonly employed in medications, agricultural products, and food preservation agents, because of their high volatility, biodegradability, and ephemeral characteristics [9]. 

Chemical type is used to describe the differences, diversity, and complexity of secondary metabolites between individuals or even populations. The main components of TEOs belong to the chemical classes of terpenoids, terpene alcohols, phenolic derivatives, ketones, aldehydes, ethers, and esters [10]. Due to the large diversity in the varieties and complex ecological environment, coupled with frequent hybridization among different populations, the chemical types of thyme have become intricate and diverse. Among them, the most common chemical types are phenols, including thymol-, carvacrol-type, etc., as well as linalool-, borneol-, eucalyptol-, and some mixed types [11,12]. The research results of Kim et al. [13] showed that ‘Lemon’, ‘Silver’, and ‘Odae’ belong to the thymol-type, with thymol contents of 43.91%, 66.24%, and 30.54%, respectively. ‘Creeping’, ‘Golden’, ‘Orange’, and ‘Wolchul’ belong to the geraniol-type, with geraniol contents of 29.57%, 65.99%, 44.70%, and 42.94%, respectively. ‘Carpet’ and ‘Jiri’ belong to the linalool-type, with linalool contents of 48.16% and 47.89%, respectively. In addition, Trendafilova et al. [14] found new chemotypes of *T. atticus* (caryophyllene oxide/β-caryophyllene), *T. leucotrichus* (β-caryophyllene/elemol/germacrene D), and *T. striatus* (β-caryophyllene/germacrene D/caryophyllene oxide).

The classification of chemical types is usually determined by the proportion of components in the essential oils [15]. And the proportion of the components in an essential oil determines the function of essential oils. However, the wide range of chemical compounds display various biological activity because of their different chemical characteristics. In addition, research has shown that microbial food spoilage is responsible for about 25% of food losses [16]. And foodborne diseases caused by pathogenic bacteria are a major public health problem worldwide. *Escherichia coli* (*E. coli*) and *Staphylococcus aureus* (*S. aureus*) are common pathogenic bacteria present in food-processing environments and are responsible for a substantial proportion of foodborne diseases [17,18]. Previous studies have shown that thymol can help maintain quality and reduce the decay of fruits and vegetables by inhibiting microbial growth during postharvest storage, where the antimicrobial effect has been mainly linked to the reduction in ergosterol, resulting in disruption of the cell membrane integrity of microorganisms [19,20,21]. Carvacrol is a biocidal product leading to bacterial membrane perturbations; in turn, this can result in the leakage of intracellular ATP and potassium ions and ultimately cell death [22]. Each active cell functions as a miniature factory, continually engaging in numerous chemical reactions. During these chemical processes, free radicals inevitably emerge, potentially causing varying degrees of damage to the cells. Antioxidants function by diminishing the activity of their structure, facilitating the neutralization of free radicals, breaking down peroxides, and chelating transition-metal ions [23,24,25]. Brewer [26] found that antioxidant activity is influenced by the aromatic rings, as well as the arrangement and quantity of hydroxyl groups. Thymol, γ-terpinene, and p-cymene exhibit potent antioxidant and antibacterial effects, as well as the ability to reduce cellular glucose intake and block lactate synthesis [2,3,4]. 

The composition of essential oils is complex, containing dozens or even hundreds of compounds. Integrated multivariate analyses have proven effective in characterizing the chemical profiles of essential oils by mitigating redundant information [27]. For example, principal component analysis (PCA) is a multivariate statistic method based on a covariance matrix between linear combinations of experimental variables, and PCA is useful for data reduction and identification of the characteristic compounds of EOs.

In this study, we used various thyme samples collected from European and Chinese native species to extract essential oils, we categorized these samples chemically by measuring the volatile components using GC/MS. Then, we assessed the chemical profiles of eight TEOs from various thyme species using a multidimensional assessment method including correlation, dendrogram, and principal component analysis (PCA). The antioxidant activity of the eight TEOs were investigated using a DPPH free radical scavenging test, and three standards of main chemical type were used to evaluate the antibacterial and antimigratory activity for A549 cells. Overall, our results provide important insights into the chemical characteristics and potential biological functions of TEO, and address their future uses related to antioxidant activity and biological effects.

## 2. Results

### 2.1. Phenotypic Evaluation of Different Thyme Species 

According to the growth habits of thyme species, the plant can be divided into two types: erect- and creeping-type. These two types exhibit striking morphological distinctions. Figure 1 displays the plant forms of eight distinct thyme species. Among these species, *Thymus vulgaris* ‘Fausitinoi’ (TvF), *Thymus pulegioides* (Tp), *Thymus rotundifolia* (Tr), *Thymus thracicus* (Tt), *Thymus longicaulis* (Tl) are erect-type, while *Thymus vulgaris* (Tv), *Thymus quinquecostatus* (Tq), and *Thymus serpyllum* (Ts) belong to the creeping-type. In addition to the differences in plant type, leaf color, leaf type, flower color, and pattern are different. The flower colors of *T. pulegioides*, *T. rotundifolia*, *T. longicaulis*, and *T. quinquecostatus* are pink, the flower colors of *T. vulgaris* ‘Fausitinoi’, *T. thracicus*, and *T. vulgaris* are white to light pink. The leaf colors of *T. vulgaris* ‘Fausitinoi’, *T. rotundifolia*, *T. thracicus*, are *T. vulgaris* are dark green and leaf blade shape is ovate, while the leaves of *T. pulegioides* are oil-green and triangular in shape. *T. longicaulis* and *T. quinquecostatus* are green in color and oval in shape. The leaf color of *T. serpyllum* is yellow green. And the flowering seasons of the eight thymes are at different times. *Thymus longicaulis* blooms earlier, the flowering season being from late April to late May, while the other thymes bloom from early July to early August.

### 2.2. Essential Oil Components from Eight Thyme Species

According to Table 1, the yields of TEOs ranged from 0.53% to 1.63% dry matter (*v*/*w*). Tt had the highest yield (1.63%) of all the samples, with thymol (51.68%) being the most dominant. Other important components included p-cymene (19.22%), γ-terpinene (10.51%), linalool (3.25%), caryophyllene (1.76%), terpinolene (1.62%), β-myrcene (1.50%), and α-thujene (1.21%). The EO yields of Tr and Ts were more than 1.5%, and the main chemical components were thymol (42.75% and 47.31%, respectively), p-cymene (21.81% and 21.15%, respectively), and γ-terpinene (11.66% and 11.16%, respectively). Tp had the lowest EO yield (0.53%) of all the samples, with thymol (48.32%), carvacrol (20.61%), and p-cymene (8.70%) being the top three components in this species. Overall, the TEO yields decreased in the following order: Tt > Tr > Ts > Tl > TvF > Tv > Tq > Tp.

The main components in the eight essential oils of thyme were determined using GC-MS. Statistical analysis was conducted on 25 components (these compounds showed relative contents greater than 0.01%) for 91.54–96.18% of the total TEO components, and the results are shown in Table 1. From the data in Table 1, we found that the compositions of TEOs varied greatly among the eight species. Furthermore, the details of these differences were visualized in a heatmap. As shown in Figure 2a and Table 1, the chemical components of the TEOs were mainly terpenoids, with a relative percentage of 90.25–95.06%, including monoterpene hydrocarbons (0–41.94%), oxygenated monoterpenes (49.42–92.23%), sesquiterpene hydrocarbons (1.64–10.11%), and oxygenated sesquiterpenes (0–0.61%). Among them, oxygenated monoterpenes were the main components of the TEOs, followed by monoterpene hydrocarbons, and the two components had a relatively large variation. The contents of sesquiterpene hydrocarbons and oxygenated sesquiterpenes were relatively low in most species detected, especially oxygenated sesquiterpenes, which were almost undetectable. In monoterpene hydrocarbons, p-cymene and γ-terpinene were the main components, showing high percentages in TvF (25.60% and 7.74%, respectively), Tr (21.81% and 11.66%, respectively), Tt (19.22% and 10.51%, respectively), Ts (21.15% and 11.16%, respectively), and Tv (17.25% and 17.80%, respectively). In oxygenated monoterpenes, geraniol, thymol, carvacrol, and nerol acetate were the main components, showing high percentages in TvF (thymol, 45.74%; carvacrol, 4.43%), Tp (thymol, 48.32%; carvacrol, 20.61%), Tr (thymol, 42.75%; carvacrol, 2.18%), Tt (thymol, 51.68%), Tl (geraniol, 37.89%; nerol acetate, 41.02%), Tq (geraniol, 74.04%; thymol, 1.36%; nerol acetate, 1.36%), Ts (thymol, 47.31%; carvacrol, 3.60%), and Tv (thymol, 42.44%; carvacrol, 2.28%). In six of the eight TEOs tested, the percentage of thymol exceeded 40%. In addition, it was found that carvacrol was 20.61% in Tp essential oil. Interestingly, we observed a high percentage of nerol acetate (41.02%) in the EOs of Tl, with both thymol and carvacrol absent. Only three sesquiterpene hydrocarbons were detected, showing low percentages in TvF (caryophyllene, 1.64%), Tp (caryophyllene, 1.46%; β-bisabolene, 2.46%), Tr (caryophyllene, 1.56%; β-bisabolene, 1.26%), Tt (caryophyllene, 1.76%), Tl (caryophyllene, 7.30%; germacrene D, 0.63%; β-bisabolene, 2.18%), Tq (caryophyllene, 7.30%; germacrene D, 0.63%; β-bisabolene, 2.18%), Ts (caryophyllene, 1.80%; germacrene D, 0.66%), and Tv (caryophyllene, 2.50%). And among the three sesquiterpenes, caryophyllene had the highest percentage content. Only caryophyllene oxide was detected in oxygenated sesquiterpenes, and the highest percentage was found in Tp (0.5%).

### 2.3. Multivariate Statistical Analysis of Essential Oil Components from Eight Thyme Species

The primary components present in the essential oils of the eight species of thyme were subjected to principal component analysis (PCA) analysis. The results showed that the first PC axis and the secondary PC axis explained 94.10% of the overall variance (Figure 2b). Tl and Tq were clearly separated from Tt, Tr, Ts, TvF, Tv, and Tp according to PC1; and Tl and Tq were clearly separated according to PC2. In conclusion, the essential oil component variation between Tt, Tr, Ts, TvF, Tv, and Tp was obviously high. Dendrogram analysis confirmed the results of the PCA analysis, in which the 25 main volatile components from TEOs were clustered into 3 different clusters (Figure 2c). The first cluster was mainly composed of thymol, and Tt, Tr, Ts, TvF, Tv, and Tp belonged to this group, which can be defined as thymol-type; Tq was clustered into a second cluster because its main component was geraniol, defined as geraniol-type; Tl was clustered into a third cluster because of its main component was nerol acetate, defined as nerol acetate-type. In terms of thyme species correlations, the strongest positive correlation was observed in the first cluster, the *p* of which all exceeding 0.8, while all samples from cluster1 were almost unrelated to the samples from the cluster2 and cluster3. In addition, the correlation between the cluster2 samples and the cluster3 samples was approximately 0.67 (Figure 2d).

### 2.4. Characterization of Shared and Unique TEO Components from Eight Thyme Species

By analyzing the shared and unique components of eight TEOs using an UpSet plot, the results were as shown in Figure 3a. The findings revealed that the number of components present in each TEO ranged from 10 to 16. Furthermore, we found that the majority of components were shared by more than one cultivar. For example, p-cymene, γ-terpinene, linalool, terpinen-4-ol, thymol, and caryophyllene were present in seven samples. Of these, only caryophyllene was shared in all samples. Four components were unique, including α-pinene, which was unique in Tv; camphene, which was unique in Tq; carvacrol methylether, which was unique in Tr; and bornyl acetate, which was unique in Tl. The chemical profiles of the p-cymene, γ-terpinene, linalool, terpinen-4-ol, thymol, and caryophyllene shared TEO components are presented in Figure 3b. Tt, Tr, Ts, TvF, Tv, and Tp showed the highest p-cymene (19.22%, 21.81%, 21.15%, 25.6%, 17.25% and 8.7%, respectively) and thymol (51.68%, 42.75%, 47.31%, 45.74%, 42.44%, and 48.32%, respectively). In the EO of Tq, although all six compounds were contained, the contents of six compounds were low. These results suggested that the composition of Tq essential oil was quite different from that of the other six TEOs, but the types of compounds were similar.

### 2.5. Variation Analysis of TEO Components from Eight Thyme Species

Given the above analysis results, we used each compound in the EOs as a categorical variable and the proportion of compound content as a continuous variable to analyze the degree of variation in 25 main volatile components. The results from the variation analysis of TEO components from eight thyme species are presented in Figure 4. The percentage contents of 22 components in the 25 main volatile components mainly varied between 0 and 10%. It was noted that geraniol was the most variable of all components (0–74.04%), followed by thymol (0–51.68%) and nerol acetate (0–41.02%), and then p-cymene (0–25.60%). Of these, thymol was mainly concentrated between 40 and 50%, while p-cymene was mainly concentrated at 20%.

### 2.6. Evaluation of the Antioxidant Activity of Eight Thyme Species

The DPPH free radical scavenging results showed that all eight essential oils had significant antioxidant activity in a dose-dependent manner, and the antioxidant activity increased with the increase in essential oil concentration, as shown in Figure 5a. When the concentration of essential oils was 0.05 mg/mL, the antioxidant activity of the essential oils was the weakest, and when the concentration of essential oils was 1 mg/mL, the antioxidant activity of the essential oils was the strongest. In addition, when the concentration of essential oils was 0.5 mg/mL and 0.6 mg/mL, there was no significant difference in the free radical scavenging rates of the different kinds of thyme; in TvF and Tv, the free radical scavenging rates were nearly the same. Three of eight TEOs had stronger antioxidant activity than vitamin E at a concentration of 1 mg/mL (Tp, 1.26– fold; Tt, 1.17– fold; Ts, 1.07– fold). Furthermore, in this work, we found that the eight essential oils ranked in order of antioxidant activity as follows: Tp > Tt > Ts > TvF > Tr > Tv > Tl > Tq.

Then, we analyzed the volatile components and DPPH activity of thyme in conjunction (Figure 5b). A total of 16 compounds were positively correlated with DPPH scavenging ability, which were ranked in descending order of correlation: thymol, carvacrol, p-cymene, 1-octen-3-ol, D-limonene, α-thujene, caryophyllene oxide, β-bisabolene, γ-terpinene, eucalyptol, thymol methyl ether, linalool, terpinen-4-ol, β-myrcene, and terpinolene. There were nine compounds negatively correlated with DPPH scavenging ability, which were ranked in descending order of correlation, as follows: geraniol, endo-borneol, germacrene D, camphene, caryophyllene, and α-pinene. In conclusion, thymol and carvacrol played a key role in the antioxidant activity, while geraniol and nerol acetate had a weak antioxidant activity, which further indicated that the difference in antioxidant capacity of these eight essential oils may be due to the difference in the proportion of monoterpenoids and that thymol/carvacrol-type essential oils had the best antioxidant activity, followed by geraniol/nerol acetate-type, while geraniol-type essential oil had a relatively weak antioxidant activity.

### 2.7. Other Potential Biological Functions of TEO

Given the above analysis results, standards of thymol, geraniol, and nerol acetate representing the chemical type of TEOs from eight thyme species were used to evaluate potential roles in antibacterial and tumor therapy. The results showed that the antibacterial activity of thymol against *E. coli* and *S. aureus* was better than geraniol and nerol acetate, and nerol acetate had no significant inhibition of either *E. coli* and *S. aureus*. Furthermore, *S. aureus* was more sensitive to thymol than *E. coli,* and the diameter of the inhibitory zone (DIZ) was measured at 48.23 mm (Figure 6a,b). In addition, cancer metastasis represents one of the most aggressive aspects of tumor development, and cell migration is recognized as a critical step in the metastasis process. To evaluate the effects of thymol, geraniol, and nerol acetate on A549 cells, a Boyden’s chamber migration assay was performed. The results showed that the number of migrated cells after treatment with the main components from essential oils was lower than in the control group (CK). Among the three main components, geraniol demonstrated the weakest performance, while the thymol group performed well (Figure 6c,d).

## 3. Discussion

The main chemical categories of components in TEOs include monoterpene hydrocarbons, oxygenated monoterpenes, sesquiterpene hydrocarbons, oxygenated sesquiterpenes, ketones, aldehydes, ethers, and esters [10,28]. Among them, thymol and carvacrol are phenolic derivatives and the most common chemical types in TEOs [29,30]. In this study, six out of eight TEOs were thymol-type. In addition, some chemotypes are rare; for example, the chemotypes of α-terpenyl acetate (accumulated to 50–70% of α-terpenyl acetate) and cis-sabinene hydrate (accumulated to 63.2% of cis-sabinene hydrate) were detected in Lithuania and Denmark, respectively [31,32]. Previous studies have shown that the qualitative composition of TEOs is controlled genetically [33]. However, different environmental conditions (climate, soil chemistry), types of pest control, fertilization methods, extraction methods, and harvesting times can influence the quantitative composition of essential oils, as well as the percentage composition [34,35,36]. For instance, the contents of carvacrol or thymol increase with a change in temperature, whereas cis-sabinene hydrate is not influenced by temperature [36]. In *Zataria multiflora*, increasing calcium contents in soil tends to increase essential oil yield and carvacrol contents but decrease linalool contents [37]. Oregano can synthesized more thymol than other chemical components at high altitudes [38]. In bergamot fruit, the composition of the main chemical components of essential oil changed dynamically during ripening [39]. In lavender, the contents and chemical composition of essential oil were influenced by flower development, temperature, and gene expression [40]. In these eight thyme species cultivated in the same conditions, *T. thracicus* had the highest thymol contents and *T. pulegioides* had the highest carvacrol contents, but there was no thymol or carvacrol in *T. longicaulis*. This may also relate to the terpenoid metabolic pathways; therefore, different chemical types of thyme can be used as research materials for different biosynthetic pathways. Over a long period of time, several of the genes responsible for volatile synthesis may be impacted by internal genetic factors connected to anatomical and physiological characteristics of the plants and the biosynthetic pathways of the volatiles. The same plant species can have different ecotypes or chemotypes as a result of these circumstances [41].

In our investigation, there were significant variations in the abundances of numerous components among the eight tested TEOs, which is similar to findings from the oregano study [42]. However, the yields of oregano essential oils exhibited a positive correlation with the content of carvacrol/thymol. According to Lukas et al. [43], the active cymyl pathway typically results in the formation of significant amounts of phenolic monoterpenoids (mostly carvacrol and/or thymol), while concurrently maintaining high EO yields. However, this phenomenon was not found in this study. Characterization of the shared and unique TEO components from eight thyme species showed six shared components among seven of the eight thyme species. In addition, β-caryophyllene was shared by all thyme species tested. Previous studies have shown that the formation of shared components may be due to these species encountering the same selection factors, such as antimicrobial, antioxidant, antiparasitic, and pro-pollinator effects [44,45]. Our analysis of the common components provided important insights into the shared features of thyme species. 

Numerous studies have shown that several phytochemicals have antibacterial activity against various foodborne pathogens. For example, studies have demonstrated that plant compounds such as carvacrol and thymol, found in essential oils extracted from the leaf or flower, have high antibacterial activity against *Salmonella enteritidis*, *E. coli*, and *L. monocytogenes* [6,46]. In the study of Chen et al. [47], the bacteriostatic activity of thymol against *S. aureus* strains was better than that for *E. coli* strains, which may be due to the different structures of their cell walls. The cell wall of Gram-positive bacteria consists of peptidoglycans, while that of Gram-negative bacteria is more complex, comprised of a thinner peptidoglycan layer and outer phospholipid membrane. In this work, the antibacterial activity of thymol against *E. coli* and *S. aureus* was better than geraniol and nerol acetate. Furthermore, *S. aureus* was more sensitive to thymol and nerol acetate than *E. coli*. This outcome may be attributed to the fact that thymol and nerol acetate can more easily penetrate the peptidoglycan layer compared to the phospholipid membrane. However, a major limitation of EO (such as thymol, carvacrol, geraniol and nerol acetate) application is their low water solubility, high volatility, and strong aromatic odor. To date, some EO hybrid composites have been proposed as a technology to overcome this limitation, which will provide us with greater possibilities for application in food preservation, functional food, and medicinal formulations [47,48].

Each active cell functions as a miniature factory, continually engaging in numerous chemical reactions. During these chemical processes, free radicals inevitably emerge, potentially causing varying degrees of damage to cells. Antioxidants function by diminishing the activity of their structure, facilitating the neutralization of free radicals, breaking down peroxides, and chelating transition-metal ions [23,24,25]. Brewer [26] found that the aromatic rings and the arrangement and number of hydroxyl groups affected antioxidant activity. Previous studies have shown that thymol and carvacrol have strong antioxidant effects [49,50,51]. In this study, it was found that eight TEOs had an antioxidant capacity, and three of them had stronger antioxidant capacity than vitamin E at a concentration of 1 mg/mL. Furthermore, it was found that the antioxidant capacity of eight TEOs was related to the sum of thymol and carvacrol; for example, Tp had the strongest antioxidant capacity and the highest contents of thymol and carvacrol (68.92%); the antioxidant ability of Tt was second, and the sum of its thymol and carvacrol contents was 51.68%; Tq had the worst antioxidant ability, and the contents of carvacrol and thymol in Tq was the lowest (1.36%). The correlation analysis showed that thymol and carvacrol had a significant contribution to the antioxidant ability in this method, indicating that TEOs containing high levels of thymol and carvacrol have stronger antioxidant capacity. These results were similar to those in previous studies [6]. 

## 4. Materials and Methods

### 4.1. Plant Materials

The eight thyme materials were obtained from the Institute of Botany, Chinese Academy of Sciences (IB-CAS), Beijing, China (Figure 1). *T. vulgaris* ‘Fausitinoi’, *T. pulegioides*, *T. rotundifolia*, *T. thracicus*, *T. longicaulis*, *T. serpyllum*, and *T. vulgaris* originate from Europe, while *T. quinquecostatus* is a Chinese native thyme. The thyme materials were two-year-old seedlings under normal water and fertilizer management. All the aerial parts of the plants at the full bloom stage were collected in 2019 and dried in the shade (20–25 °C) for further analyses. Each sample was composed of at least 6 plants.

### 4.2. Extraction of Essential Oils

Each dried thyme sample was crushed into a powder. The 100 g of powdered samples was combined with 1000 mL of distilled water to extract the essential oils through steam distillation. After boiling, the extraction procedure was carried out for three hours. The essential oils were extracted, dried with anhydrous sodium sulfate, and kept at 4 °C in an amber bottle. The yields were calculated as the dry weight of plant materials (in% *v*/*w* of 100 g dried raw material).

### 4.3. Gas Chromatography Mass Spectrometry Analysis of Essential Oils

The essential oils were filtered via a filter membrane of 0.22 μm and diluted with n-hexane. A 7890A-7000B GC-MS (Agilent Technologies, Santa Clara, CA, USA) outfitted with an HP-5MS column (30 m × 0.25 mm × 0.25 μm; Agilent Technologies) was then used to conduct gas chromatography mass spectrometry (GC-MS). The temperature of the injector was 280 °C. The oven program had the following settings: the temperature was kept at 40 °C for 2 min, then increased linearly to 260 °C at a rate of 4° C/min; and finally, the temperature was increased to 310 °C at a rate of 60 °C/min. The following are the MS settings: quadrupole, electronic impaction source temperature of 230 °C, ionization energy of 70 eV, and 50–500 u of mass range.

The RI values were determined using n-alkane hydrocarbons (C7-C40, Sigma, Burbank, CA, USA) under the same conditions. The relative percentage of essential oil components was determined based on the peak area. Retention index (RI) values and the spectra from the 17.0 library of the National Institute of Standards and Technology were compared to identify the compounds [52]. Under the same circumstances, the RI values were calculated using n-alkane (C7-C40) hydrocarbons. 

### 4.4. In Vitro Antioxidant Activity

Using a previously described approach with some modifications, the 2,2-diphenyl-1-picrylhydrazyl (DPPH) scavenging activity of extracts from eight essential oils was evaluated [53]. Briefly, 150 μL of an ethanol solution of DPPH (0.1 mM) was mixed with 50 μL of a solution of essential oils (0.05 mg/mL, 0.1 mg/mL, 0.2 mg/mL, 0.4 mg/mL, 0.5 mg/mL, 0.6 mg/mL, 0.8 mg/mL, and 1 mg/mL, respectively) and incubated for 30 min in the dark, before the absorbance at 517 nm was measured. In addition, 1 mg/mL of vitamin E was used to as a positive control. We accurately weighed 5 mg of essential oil or vitamin E and dissolved them in 100 mL, 50 mL, 25 mL, 12.5 mL, 10 mL, 8.33 mL, 6.25 mL, and 5 mL of ethanol solution to obtain 0.05 mg/mL, 0.1 mg/mL, 0.2 mg/mL, 0.4 mg/mL, 0.5 mg/mL, 0.6 mg/mL, 0.8 mg/mL, and 1 mg/mL, respectively.

### 4.5. Antibacterial Activity of the Main Components against E. coli and S. aureus

*E. coli* ATCC 25922 and *S. aureus* ATCC 25923 were obtained from the Key Laboratory of Plant Resources, Institute of Botany, Chinese Academy of Sciences. The disc diffusion method [54], with some modifications, was used to compare the zone of inhibition of thymol, geraniol, and nerolyl acetate. Briefly, 100 µL *E. coli* and *S. aureus* suspensions (approximately 10^7–8^ CFU/mL) were evenly spread onto LB agar plates. Sterilized antimicrobial disks were placed on the test plates. A 6 µL aliquot of monomer component was added to the 6 mm disc, which was incubated at 37 °C for 24 h. The diameter of the inhibitory zone (DIZ) value was measured using Vernier calipers (Airaj, Tsingtao, China). All experiments were performed in triplicate.

### 4.6. Cell Migration Ability Test

The lung adenocarcinoma cell line A549 was purchased from the Cell Resources Center, Institute of Basic Medicine, Chinese Academy of Medical Sciences. The cells were inoculated at a density of 2.5 ×10 ^5^ cells/mL. First, 700 μL medium containing 10% serum was added to the lower chamber (bottom of 24-well plate) of the matrix rubber plate, and 100 μL cells were added to the upper chamber, incubated in an incubator at 37 °C and 5% CO_2_ for 4 h, treated with 100 μL fluorouracil (5-FU), thymol, geraniol, and nerol acetate (10 µg/mL), and then cultured for 24 h. Samples were taken out of the chamber, the liquid in the lower chamber was carefully absorbed, and they were moved to a hole with 4% paraformaldehyde in advance for 10 min at room temperature. Then, the chamber was cleaned twice with PBS and moved to 100% methanol at room temperature for 20 min. The chamber was then transferred to a crystal violet solution and stained at room temperature for 15 min. After that, the chamber was cleaned with PBS, and the cells on the membrane surface at the bottom of the lower chamber were carefully wiped with a wet cotton swab. Finally, the cells were allowed to dry thoroughly and then counted and photographed under an inverted microscope 10× objective.

### 4.7. Statistical Analysis

The statistical analysis was performed with office 2010 and Origin 2021. Values are represented as the mean ± SEM. All experiments were independently performed in triplicate. Asterisk (*) indicates significant differences (** *p* < 0.01, *** *p* < 0.001, **** *p* < 0.0001). The percentage content of EOs was used to plot a heatmap, PCA score plot, and upset and raincloud plots using online software (https://www.chiplot.online, accessed on 11 November 2023). The content of each compound from the eight EOs and the DPPH scavenging activity of the extracts were used to calculate Pearson correlation coefficient with Origin 2021, and then online software (https://www.omicstudio.cn/tool/64, accessed on 6 March 2023) was used to draw a correlation network plot.

## 5. Conclusions

This study established a multidimensional analytical method to evaluate differences in the chemical profiles and potential biological functions of EOs from eight thyme species. The results showed that the TEOs exhibited chemical diversity, and three main clusters were identified (thymol-, geraniol-, and nerol acetate-type) and thymol was the main type. The percentage contents of 22 components in the 25 main volatile components were mainly concentrated in 0–10%. Geraniol was the most variable of all components (0–74.04%), followed by thymol (0–51.68%), nerol acetate (0–41.02%), and then p-cymene (0–25.60%). In addition, eight TEOs had some common compounds, such as thymol and p-cymene, which may encounter the same selection factors. The eight TEOs had antioxidant activity, and the contents of thymol and carvacrol were positively correlated with the antioxidant activity of the TEOs. The antibacterial and tumor cell migration inhibition experiments with three standards also indicated that the thymol-type EOs have great potential for antibacterial and tumor inhibition. Overall, our results contributed to the understanding of these TEOs and can provide a theoretical basis for further exploration of the function of natural products from thyme essential oils.

## Figures and Tables

**Figure 1 plants-12-04164-f001:**
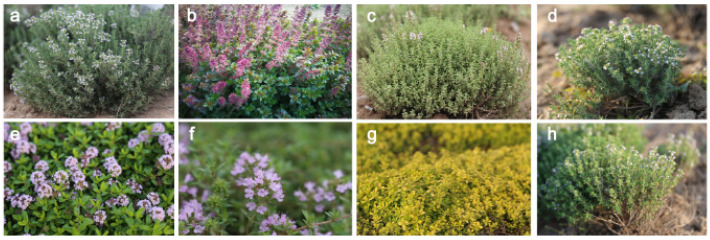
Plant morphology of eight thymes. (**a**) *Thymus vulgaris* ‘Fausitinoi’; (**b**) *Thymus pulegioides*; (**c**) *Thymus rotundifolia*; (**d**) *Thymus thracicus*; (**e**) *Thymus longicaulis*; (**f**) *Thymus quinquecostatus*; (**g**) *Thymus serpyllum*; (**h**) *Thymus vulgaris*.

**Figure 2 plants-12-04164-f002:**
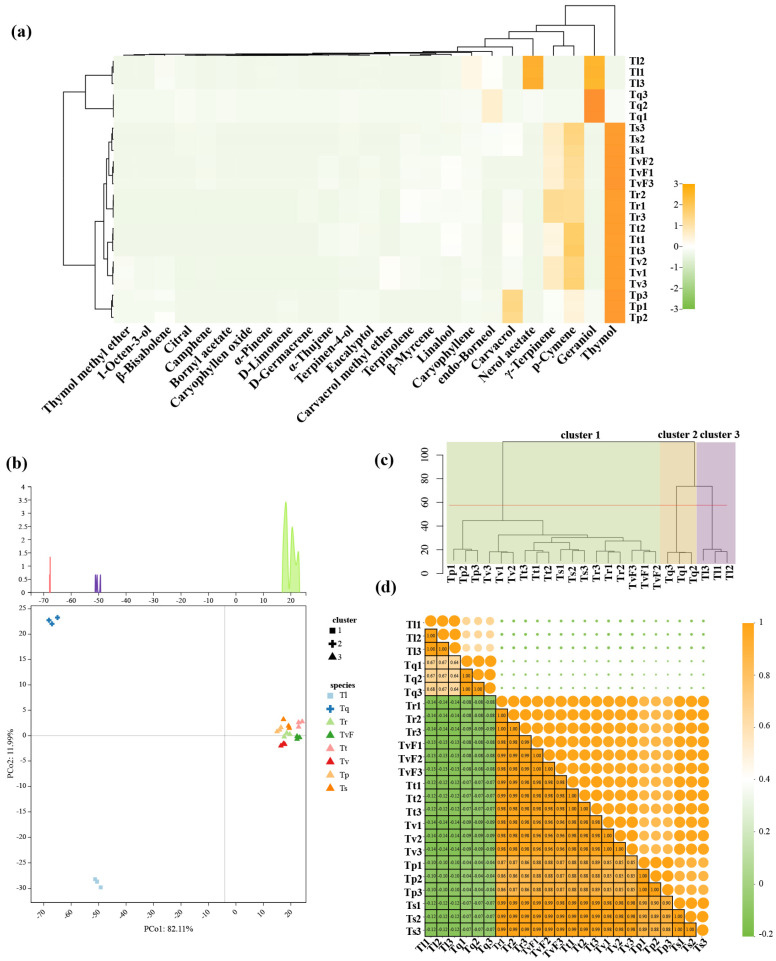
Analysis of volatile components of eight thyme essential oils. (**a**) Heatmap; (**b**) PCA score plot; (**c**) dendrogram; (**d**) Pearson correlation plot. TvF, *T. vulgaris* ‘Fausitinoi’; Tp, *T. pulegioides*; Tr, *T. rotundifolia*; Tt, *T. thracicus*; Tl, *T. longicaulis*; Tq, *T. quinquecostatus*; Ts, *T. serpyllum*; Tv, *T. vulgaris*.

**Figure 3 plants-12-04164-f003:**
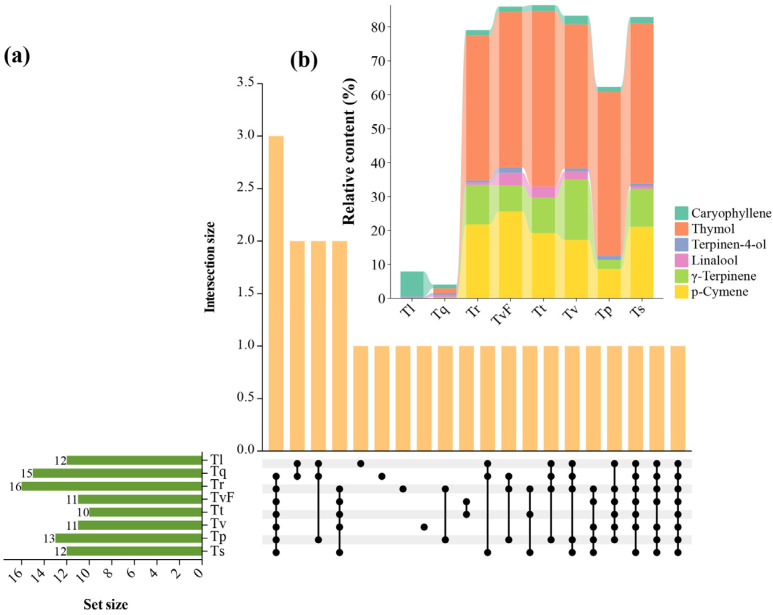
Distributions of shared and unique components from eight TEOs. (**a**) UpSet plot represents shared and unique components from eight thyme essential oils; (**b**) the percentage of the main shared chemical composition of thyme essential oils. TvF, *T. vulgaris* ‘Fausitinoi’; Tp, *T. pulegioides*; Tr, *T. rotundifolia*; Tt, *T. thracicus*; Tl, *T. longicaulis*; Tq, *T. quinquecostatus*; Ts, *T. serpyllum*; Tv, *T. vulgaris*.

**Figure 4 plants-12-04164-f004:**
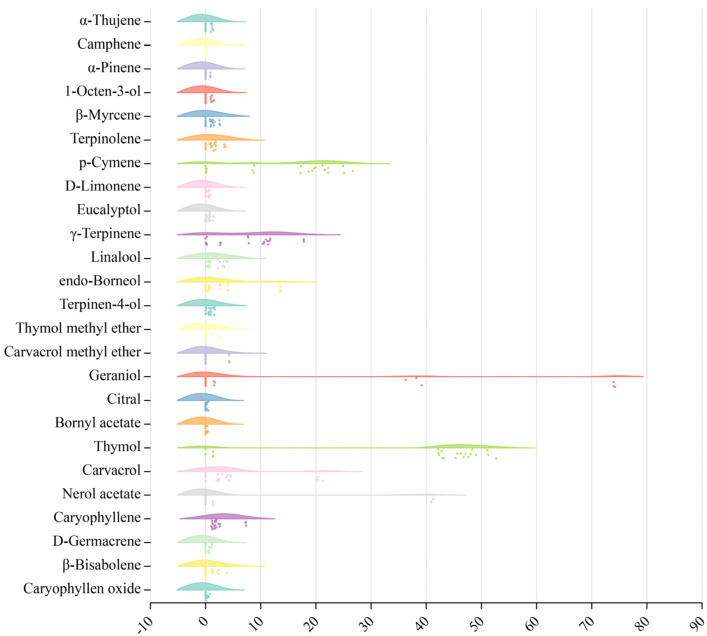
Variation analysis of TEO components from eight thyme species. The percentage content of EOs was used to plot a raincloud plot using online software (https://www.chiplot.online, accessed on 14 October 2023).

**Figure 5 plants-12-04164-f005:**
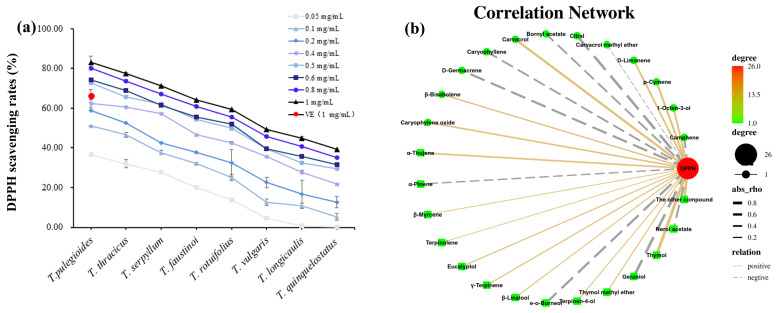
Antioxidant activity of essential oils from eight thyme species. (**a**) DPPH scavenging activity, VE: vitamin E (1 mg/mL); (**b**) correlation analysis of volatile components and DPPH activity.

**Figure 6 plants-12-04164-f006:**
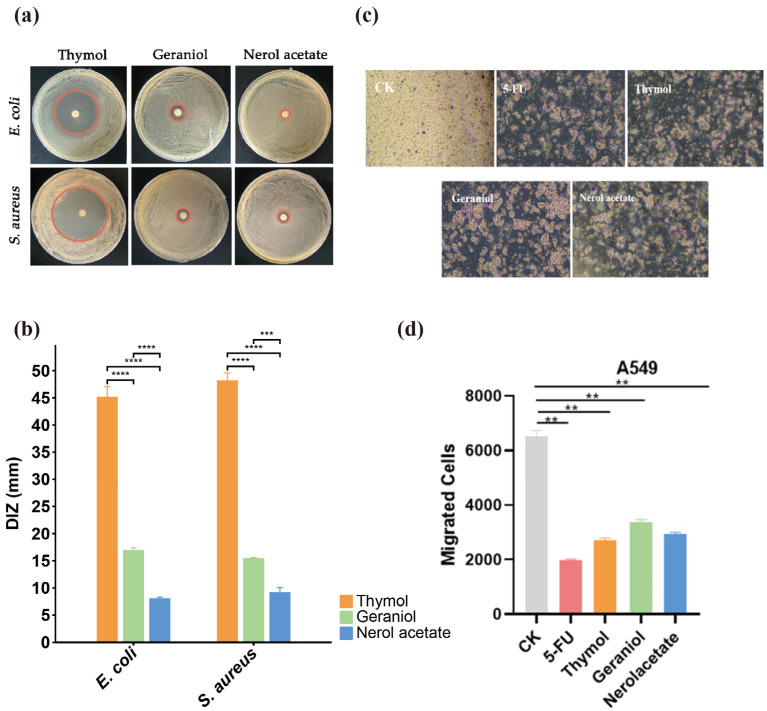
Potential biological functions of three chemical type standards in antibacterial and tumor therapy. (**a**) Diameter of the inhibitory zone (DIZ) images; (**b**) statistical data of DIZ; (**c**,**d**) effects of three chemical type standards on A549 cell migration. CK: PBS treatment; 5-FU: fluorouracil, positive control. The 100 uL 5-FU of thymol, geraniol, and nerol acetate was applied at a concentration 10 µg/mL. Asterisk (*) indicates significant differences: ** *p* < 0.01, *** *p* < 0.001, **** *p* < 0.0001.

**Table 1 plants-12-04164-t001:** The chemical composition of thyme essential oils from eight species.

Compound	RI ^a^	RT ^b^	Relative Concentration (%)							
			TvF	Tp	Tr	Tt	Tl	Tq	Ts	Tv
α-Thujene	920	921	1.06 ± 0.13	-	-	1.21 ± 0.12	-	-	-	-
α-Pinene	929	921	-	-	-	-	-	-	-	0.87 ± 0.01
Camphene	952	936	-	-	-	-	-	0.21 ± 0.02	-	-
1-Octen-3-ol	980	968	-	1.30 ± 0.20	0.98 ± 0.04	-	-	-	-	-
β-Myrcene	991	982	1.28 ± 0.04	-	1.08 ± 0.11	1.50 ± 0.14	-	-	0.89 ± 0.04	2.55 ± 0.07
Terpinolene	1017	1008	0.98 ± 0.02	-	1.80 ± 0.06	1.62 ± 0.31	-	-	1.27 ± 0.50	3.48 ± 0.15
p-Cymene	1025	1018	25.60 ± 0.98	8.70 ± 0.16	21.81 ± 0.36	19.22 ± 0.50	-	0.17 ± 0.01	21.15 ± 1.1	17.25 ± 0.06
D-Limonene	1030	1021	-	-	0.44 ± 0.10	0.66 ± 0.11	-	-	0.63 ± 0.08	-
Eucalyptol	1032	1030	0.76 ± 0.06	0.82 ± 0.01	0.63 ± 0.08	0.66 ± 0.06	0.15 ± 0.02	-	-	1.43 ± 0.02
γ-Terpinene	1060	1058	7.74 ± 0.08	2.68 ± 0.07	11.66 ± 0.08	10.51 ± 0.14	-	0.21 ± 0.02	11.16 ± 0.20	17.80 ± 0.02
Linalool	1099	1097	3.64 ± 0.38	-	0.64 ± 0.01	3.25 ± 0.12	0.43 ± 0.01	0.61 ± 0.01	0.83 ± 0.08	2.34 ± 0.24
endo-Borneol	1167	1164	-	-	0.51 ± 0.07	-	4.06 ± 0.06	13.54 ± 0.01	2.63 ± 0.05	-
Terpinen-4-ol	1177	1176	1.63 ± 0.05	1.12 ± 0.18	0.62 ± 0.04	-	0.15 ± 0.01	0.61 ± 0.03	0.64 ± 0.08	0.93 ± 0.02
Thymol methyl ether	1235	1235	-	1.19 ± 0.13	2.34 ± 0.07	-	-	0.20 ± 0.01	-	-
Carvacrol methyl ether	1244	1244	-	-	4.29 ± 0.06	-	-	-	-	-
Citral	1247	1247	-	-	-	-	0.23 ± 0.02	0.49 ± 0.01	-	-
Geraniol	1255	1254	-	-	-	-	37.89 ± 1.48	74.04 ± 0.15	-	-
Bornyl acetate	1285	1286	-	-	-	-	0.35 ± 0.03	-	-	-
Thymol	1291	1292	45.74 ± 0.44	48.32 ± 0.70	42.75 ± 0.56	51.68 ± 0.87	-	1.36 ± 0.00	47.31 ± 0.66	42.44 ± 0.30
Carvacrol	1299	1300	4.43 ± 0.18	20.61 ± 0.55	2.18 ± 0.16	-	-	-	3.60 ± 0.52	2.28 ± 0.06
Nerol acetate	1382	1385	-	-	-	-	41.02 ± 0.17	1.36 ± 0.01	-	-
Caryophyllene	1419	1422	1.64 ± 0.08	1.46 ± 0.44	1.56 ± 0.26	1.76 ± 0.09	7.30 ± 0.08	1.13 ± 0.12	1.80 ± 0.14	2.50 ± 0.16
Germacrene D	1481	1483	-	-	-	-	0.63 ± 0.10	1.09 ± 0.01	0.66 ± 0.08	-
β-Bisabolene	1509	1511	-	2.46 ± 1.36	1.26 ± 0.03	-	2.18 ± 0.12	0.01 ± 0.01	-	-
Caryophyllene oxide	1581	1578	-	0.61 ± 0.19	-	-	0.21 ± 0.02	0.01 ± 0.01	-	-
Monoterpene hydrocarbons			36.65 ± 1.06	11.38 ± 0.20	36.78 ± 0.24	34.72 ± 0.65	-	0.59 ± 0.03	41.94 ± 0.23	35.09 ± 1.13
Oxygenated monoterpenes			56.19 ± 0.40	74.34 ± 0.40	53.96 ± 0.71	55.58 + 0.83	84.26 ± 1.32	92.23 ± 0.14	49.42 ± 0.56	55.01 ± 0.73
Sesquiterpene hydrocarbons			1.64 ± 0.08	3.92 ± 1.36	2.81 ± 0.24	1.76 ± 0.09	10.11 ± 0.06	2.24 ± 0.13	2.50 ± 0.16	2.46 ± 0.16
Oxygenated sesquiterpenes			-	0.61 ± 0.19	-	-	0.21 ± 0.01	0.01 ± 0.01	-	-
Total			96.18 ± 1.49	91.54 ± 1.27	94.19 ± 0.64	92.19 ± 0.12	94.58 ± 1.37	95.06 ± 0.31	92.56 ± 0.68	93.85 ± 0.88
EO yields (%)			1.09 ± 0.12	0.53 ± 0.06	1.60 ± 0.05	1.63 ± 0.08	1.16 ± 0.05	0.78 ± 0.03	1.50 ± 0.06	1.00 ± 0.02

The numbers are arranged in order of retention time, and values (relative peak area percent) represent averages of three determinations (“-” indicates no detection); RI ^a^: retention index of the component on semi-standard non-polar capillary column from NIST and reported literature; RI ^b^: calculated retention index of standard mixture of n-alkanes on HP-5MS capillary column. TvF, *T. vulgaris* ‘Fausitinoi’; Tp, *T. pulegioides*; Tr, *T. rotundifolia*; Tt, *T. thracicus*; Tl, *T. longicaulis*; Tq, *T. quinquecostatus*; Ts, *T. serpyllum*; Tv, *T. vulgaris*.

## Data Availability

Data are contained within the article.

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
