# Peer review of "Chemical Compositions of Essential Oil Extracted from Eight Thyme Species and Potential Biological Functions"

_plants, 2023, doi:10.3390/plants12244164_

Round 1

Reviewer 1 Report

Comments and Suggestions for Authors

Authors revised the manuscript according to comments of the three reviewers. The manuscript is acceptable for publication.

Comments on the Quality of English Language

Minor editing changes are necessary.

Author Response

Comments 1: [Authors revised the manuscript according to comments of the three reviewers. The manuscript is acceptable for publication.]

Response 1: Thank you very much for your approval of the manuscript.

Comments 2: [Minor editing changes are necessary on the Quality of English Language.]

Response 2: Thank you very much for your opinions. We have read the entire text thoroughly, corrected grammar errors, and also consulted others to proofread the language.

Reviewer 2 Report

Comments and Suggestions for Authors

The authors have responded to all the requests made by the reviewers and therefore I think it can be published in this form. 

Best regards 

Author Response

Comments 1: [The authors have responded to all the requests made by the reviewers and therefore I think it can be published in this form.]

Response 1: Thank you very much for your approval of the manuscript.

Reviewer 3 Report

Comments and Suggestions for Authors

The present reseach evaluates the composition of the essential oils of 8 cultivars of Thyme cultivated in China, as well as the antioxydant, anti bacterial and antitumural activity.

The abstract is ok

The introduction should be reviewed as the first paragraphs uses references that are not adapted to the statements cited. The objectives of the research are not clear enough and not enough justified.

In the material and method section, there are data missing mainly on the growing conditions and samples preparation.

The results are quite clear, table 1 should be placed sooner in the text, some stastistical analysis are missing in the comparison of the antioxydant potential compared to Vitamin E.

Discussion is ok

Conclusion is ok

English should be reviewed there are quite a lot of sentences which are not clear.

More detailed comments and suggestions are present in the attached document

Comments on the Quality of English Language

English exprission is not equal along the document but some sentences are not in a clear english and others are not enough complet to give an interesting content (only saying vanalities)

Author Response

Comments 1: [The abstract is ok]

Response 1: Thank you very much for your approval of the abstract.

Comments 2: [The introduction should be reviewed as the first paragraphs uses references that are not adapted to the statements cited. The objectives of the research are not clear enough and not enough justified.]

Response 2: Thank you very much for your opinions. We have made modifications to the introduction, such as deleting reference 1, replacing references 9, 18, 19, and proposing the research objectives. Please refer to the manuscript for details, all modifications are highlighted in green.

Comments 3: [In the material and method section, there are data missing mainly on the growing conditions and samples preparation.]

Response 3: Thank you very much for your opinions. We have added relevant information, such as “a solution of essential oils (0.05 mg/mL, 0.1 mg/mL, 0.2 mg/mL, 0.4 mg/mL, 0.5 mg/mL, 0.6 mg/mL, 0.8 mg/mL, 1 mg/mL, respectively)”. Please refer to the manuscript for details, all modifications are highlighted in green.

Comments 4: [The results are quite clear, table 1 should be placed sooner in the text, some stastistical analysis are missing in the comparison of the antioxydant potential compared to Vitamin E.]

Response 4: Thank you very much for your opinions. We have adjusted the order. In addition, we agree that the antioxidant potential should be analyzed statistically compared to vitamin E. However, vitamin E only had one concentration and all dates were presented in a line chart format, the labeling of significance analysis will be prone to confusion. Therefore, we have provided clear multiples in the manuscript: “And three of eight TEOs had stronger antioxidant activity than vitamin E in the concentration of 1 mg/mL (Tp, 1.26– fold; Tt, 1.17– fold; Ts, 1.07– fold).”. Because we also believe that specific numerical increases are sometimes more meaningful than significance analysis.

Comments 5: [Discussion is ok]

Response 5: Thank you very much for your approval of the Discussion.

Comments 6: [Conclusion is ok]

Response 6: Thank you very much for your approval of the Conclusion.

Comments 7: [English should be reviewed there are quite a lot of sentences which are not clear.]

Response 7: Thank you very much for your opinions. We have read the entire text thoroughly, corrected grammar errors, and also consulted others to proofread the language.

Comments 8: [when the material was harvested, which year, which period of the year, and which part(s) of the plant were collected, only leaves? stems? flowers? An equilibrate mixture. Sampling was done with one plant? several?]

Response 8: Thank you very much for your opinions. We have added relevant information: “All the aerial part of the plants at the full bloom stage were collected in 2019 and dried in the shade (20–25â—¦C) for further analyses. The per sample of each genotype/variety of thyme was harvest at least six plants.”.

Comments 9: [how you proceeded to get the material dry?]

Response 9: Thank you very much for your opinions. We have added relevant information: “dried in the shade (20–25â—¦C) for further analyses.”.

Comments 10: [this mean that you had 3 different samples of each genotype/variety of thyme and you conducted three extractions?]

Response 10: Thank you very much for your opinions. We divided the collected samples of each genotype/variety of thyme into three equal parts, and then conducted three extractions.

Comments 11: [More detailed comments and suggestions are present in the attached document]

Response 11: Thank you very much for your opinions. Please refer to the manuscript for details, all modifications are highlighted in green.

Round 2

Reviewer 3 Report

Comments and Suggestions for Authors

The authors have no provided corrections to the references in the introduction section and have improved the research objective description.

The material and method section has been improved adding the data missing in the precedent version.

The figures and tables have been adapted to the request, still in one or two in the legend is missing the entire name of the Thyme species used.

The english looks now fine.

The other sections are ok.

Some minor corrections have to be performed before publishig, you can find the exact place in the attached document

Author Response

Point-by-point response to reviewer

Comments 1: [The authors have no provided corrections to the references in the introduction section and have improved the research objective description.]

Response 1: First of all, thank you very much for your approval of the research objective description. Secondly, we have verified the references in the introduction. Because the order of all references has been changed, we did not mark them. For example, the description of relevant content was presented in reference 1: “Species of the genus Thymus are important medicinal plants that have been used in traditional medicine for thousands of years in countries of the Mediterranean basin.”

Additionally, reference 8 should have cited the European Pharmacopoeia, which has now been revised and highlighted in blue.

Comments 2: [The material and method section has been improved adding the data missing in the precedent version.]

Response 2: Thank you very much for your approval of the material and method section.

Comments 3: [The figures and tables have been adapted to the request, still in one or two in the legend is missing the entire name of the Thyme species used.]

Response 3: Thank you very much for your opinions. We have added the entire name of the Thyme species used in Figure 3: “TvF, T. vulgaris ‘Fausitinoi’; Tp, T. pulegioides; Tr, T. rotundifolia; Tt, T. thracicus; Tl, T. longicaulis; Tq, T. quinquecostatus; Ts, T. serpyllum; Tv, T. vulgaris.” and highlighted in blue.

Comments 4: [Thyme The english looks now fine.]

Response 4: Thank you very much for your approval of the English revisions.

Comments 5: [The other sections are ok.]

Response 5: Thank you very much for your approval of the manuscript.

Comments 6: [Some minor corrections have to be performed before publishig, you can find the exact place in the attached document.]

Response 6: Thank you very much for your opinions. We have completed the modifications.

Generally Recognized as Safe” has been modified to “generally recognized as safe”.

“diverse varieties” has been modified to “large diversity in the varieties or of the varieties”.

We have added the entire name of the Thyme species used in Figure 3: “TvF, T. vulgaris ‘Fausitinoi’; Tp, T. pulegioides; Tr, T. rotundifolia; Tt, T. thracicus; Tl, T. longicaulis; Tq, T. quinquecostatus; Ts, T. serpyllum; Tv, T. vulgaris.” and highlighted in blue.

“The per sample of each genotype/variety of thyme was harvest at least six plants.” has been modified to “Each sample was composed of at least 6 plants.“.
